# SCD: A Stacked Carton Dataset for Detection and Segmentation

**DOI:** 10.3390/s22103617

**Published:** 2022-05-10

**Authors:** Jinrong Yang, Shengkai Wu, Lijun Gou, Hangcheng Yu, Chenxi Lin, Jiazhuo Wang, Pan Wang, Minxuan Li, Xiaoping Li

**Affiliations:** 1State Key Laboratory of Digital Manufacturing Equipment and Technology, Huazhong University of Science and Technology, Wuhan 430074, China; yangjinrong@hust.edu.cn (J.Y.); shengkaiwu@hust.edu.cn (S.W.); getglj@hust.edu.cn (L.G.); hcy@hust.edu.cn (H.Y.); linchenxi@hust.edu.cn (C.L.); wangjiazhuo@hust.edu.cn (J.W.); panwang725@hust.edu.cn (P.W.); 2Faculty of Arts and Science, Queen’s University, Kingston, ON K7L 3N6, Canada; 18ml51@queensu.ca

**Keywords:** object detection, larger-scale dataset, stacked carton

## Abstract

Carton detection is an important technique in the automatic logistics system and can be applied to many applications such as the stacking and unstacking of cartons and the unloading of cartons in the containers. However, there is no public large-scale carton dataset for the research community to train and evaluate the carton detection models up to now, which hinders the development of carton detection. In this article, we present a large-scale carton dataset named Stacked Carton Dataset (SCD) with the goal of advancing the state-of-the-art in carton detection. Images were collected from the Internet and several warehouses, and objects were labeled for precise localization using instance mask annotation. There were a total of 250,000 instance masks from 16,136 images. Naturally, a suite of benchmarks was established with several popular detectors and instance segmentation models. In addition, we designed a carton detector based on RetinaNet by embedding our proposed Offset Prediction between the Classification and Localization module (OPCL) and the Boundary Guided Supervision module (BGS). OPCL alleviates the imbalance problem between classification and localization quality, which boosts AP by 3.1∼4.7% on SCD at the model level, while BGS guides the detector to pay more attention to the boundary information of cartons and decouple repeated carton textures at the task level. To demonstrate the generalization of OPCL for other datasets, we conducted extensive experiments on MS COCO and PASCAL VOC. The improvements in AP on MS COCO and PASCAL VOC were 1.8∼2.2% and 3.4∼4.3%, respectively.

## 1. Introduction

Carton detection is extremely important for automatic logistics transport such as robots performing stacking and unstacking operations, and especially for achieving the goal of completing the automation of logistics scenarios. Traditional methods of carton localization include manual identification, Radio Frequency Identification (RFID)-based carton position measurement [1], 3D laser scanning measurement [2], and traditional vision schemes [3]. However, these schemes always suffer from low adaptability and reliability. Recently, deep learning approaches have shown significant achievement for many tasks such as nature scene [4], face [5], defect [6], and medical object detection [7] etc. CNN-based models on visual techniques show great application prospects in carton detection or segmentation tasks under various scenarios.

The training and evaluation of an excellent task-wise deep network are strongly tied to the construction of a large-scale dataset. Building CNN-based models to detect or segment cartons requires a large scale dataset of the cartons with scenario constraints. At present, several popular and generic datasets are available for CNN-based models development, e.g., MS COCO [8], PASCAL VOC [9], ImageNet [10], and Open Image [11]. For task-oriented requirements, these are the dedicated dataset, e.g., medical tasks [12], face detection [13], aerial detection [14], etc. Nonetheless, most of them do not contain the category of carton. ImageNet and Open Image include related images, but their numbers are small. Furthermore, most of the images in these benchmarks are downloaded from the Internet, and each image only cover several instances, making it difficult to simulate the various arrangements of cartons in a stacked scenario.

To fill the shortage of carton datasets, we established a Stacked Carton Dataset (SCD) with a large scale of images only containing cartons and a wide variety of backgrounds. To further mimic different carton packing scenarios, the number of cartons in the image was arranged arbitrarily from a sparse to a dense distribution. We split the SCD into two subsets (called the Live Stacked Carton Dataset (LSCD) and the Online Stacked Carton Dataset (OSCD)) that were collected from the warehouse and downloaded from the Internet, respectively. To ensure that cartons in images have rich texture information and application occasion, different logistics and storage locations were selected to collect image data, e.g., an integrated e-commerce logistics warehouse, a logistics transfer station, a drug e-commerce logistics warehouse, and a fruit wholesale market, as well as using web tools to collect related images. The collecting spots of the online subset were also numerous to be greedily used for pre-training followed by fine-tuning a model for a specific carton scenario and performing transfer learning, etc. As the most important part of the dataset, pixel-wise annotation was carried out for more precise localization. Particularly, the outer/inner and occlusion information were provided by dividing a carton into four categories including Carton-inner-all, Carton-inner-occlusion, Carton-outer-all, and Carton-outer-occlusion. These bonus annotations have great application potential; for example, it is easy to determine which box can be grabbed by turning a logical problem into a classification problem.

SCD places unique challenges for computer vision research on detection [15,16,17]. To establish a suit of benchmark, we evaluated several CNN-based detectors including single-stage [16,17] and two-stage frameworks [15,18]. We also report the performance of recently advanced detectors like DETR [19], YOLOX [20], and CenterNet [21]. Additionally, we published a benchmark for the instance segmentation task by employing several advanced segmentation models [18,22,23,24,25]. Then, we chose RetinaNet [16] as the baseline on our dataset because of its comprehensive performance and speed. However, RetinaNet suffers from a serious disalignment [26,27] between classification and localization accuracy. In the post-processing phase of RetinaNet, the classification score of each predict bounding box is directly used to feed into the non-maximal suppression (NMS) procedure. Under this regime, the prediction boxes with higher classification scores will be retained, while the nearly lower ones will be discarded. Since the classification score can not fully reflect the positioning confidence, nearby prediction boxes that are better positioned are thrown away. To ease this situation, IoU-Net [27] and IoU-aware RetinaNet [26] were proposed to directly predict the IoU of samples. VarifocalNet [28] and generalized focal loss [29] adopted an IoU-aware classification loss to predict localization confidence, but they give up the classification confidence that is essential in some specific occasions, such as automatic pickup of goods and tumor detection. To deal with this issue, we propose Offset Prediction between the Classification and Localization module (OPCL) to bridge the gap from classification to localization quality while retaining the classification head, and the gaps were adopted to modify classification confidence.

When stacked cartons are detected, repetitive textures are often everywhere because cartons are often stacked together for the same purpose, which confuses the model to make wrong judgments. To eliminate the dilemma, we utilized the boundary information including the contour of the individual box and the topology of carton placement. Owing to the pixel-wise annotation, the boundary of each instance can be simply extracted without any other annotations. We proposed a Boundary Guided Supervision module (BGS) to supervise and urge detectors to pay more attention to boundary information, which helps decouple repeated text and does not add computation overhead during inference.

The main contributions of the study are summarized as follows:We established a large-scale Stack Carton Dataset (SCD), which contains two subsets collected from the warehouse and the Internet. Images collected on the site in different scenarios have rich texture information and high resolution, while the part of the Internet can be used for pre-training and transfer learning.Pixel-wise annotation was applied to label instances in SCD. MS COCO [8] and PASCAL VOC [9] format annotation were also provided for detection and segmentation tasks. Four labels that contain inside/outside information, and occlusion information of the carton are provided to cater to the operation requirement of carton goods. To the best of our knowledge, SCD is the first public image dataset on the carton in diverse scenarios that provides the premise for the carton cargo detection based on deep learning.We provided a benchmark including detection and instance segmentation tasks on SCD for evaluation by using several state-of-the-art deep learning models.We proposed two novel modules for RetinaNet including OPCL and BGS. OPCL was proposed to deal with the extreme imbalance of the classification score and the localization confidence with respect to the samples. BGS was s introduced to extract and utilize boundary information of cartons, which guides the model to pay more attention to boundary information so as to promote carton localization.

The article is organized as follows. Related works are discussed in Section 2. The construction details and statistics of the SCD are introduced in Section 3. The used carton detector and its novel modules are introduced in Section 4. Extensive experimental results are published in Section 5. Finally, conclusions are summarized in Section 6.

## 2. Related Work

We first review several popular datasets, and some of them contain carton images. Then, several methods for the imbalance issues between classification and localization are discussed, and finally some related works for predicting target map are introduced.

**Large-Scale Datasets.** The great development of deep neural networks is inseparable from the establishment of large datasets that provide significant annotation information for the related tasks of training models. The MS COCO [8] dataset is the most popular benchmark for evaluating novel models and poses a huge challenge to researchers. There are 80 object classes, over 1.5 million object instances, and more than 200,000 images in the COCO dataset, which supports classification, detection, and segmentation tasks. The PASCAL VOC [9] contains two versions (VOC2007 and VOC2012), which include 9963 and 22,263 images, respectively. Twenty object classes are included in the two versions and the PASCAL VOC support detection and semantic segmentation. However, both of them do not include the object class of cartons. As the most frequently used dataset for pre-training, ImageNet [10] provides image-level labels for 5247 classes and a large scale of images up to 3,200,000. This dataset has greatly prompted the development of computer vision in recent years, but only 1388 images include cartons and cannot be used for carton detection in the dense carton packing scenes. The latest Open Image dataset [11] contains 9,052,839 images and provides 7186 object classes for the classification task and 600 classes for the detection task. It includes box classes that contain carton instances, but it doesn’t subdivide into paper boxes and suffers from scarcity and inferior quality. In addition, these are several task-oriented datasets, e.g., medical image [12], face detection [13], aerial detection [14], etc., which serve specific scenarios. Our work is specially born for the carton work scene

**Misalignment of Classification and Localization Accuracy.** As the indispensable module, non-maximal suppression (NMS) plays a significant role for detectors to remove duplicated predicting bounding boxes and is used in most state-of-the-art models [15,16,17,30]. Soft-NMS [31] was proposed to adopt a decay strategy of confidence that is easy to implement and boost the recall and accuracy. Both NMS and Soft-NMS regard the classification confidence as the corresponding localization accuracy that does not give full play to the ability of regression and leads to suboptimal performance. To ease the misalignment, several works utilize an extra branch to predict various types of criterion that are integrated into the classification score to simulate the localization quality and participate in the process of removing the repeated boxes. IoU-Net [27] and IoU-Aware RetinaNet [26] directly predict the IoU between the predictive boxes and the corresponding ground truth boxes in two-stage and single-stage models, respectively. As an excellent anchor-free detector, FCOS [17] predicts centerness scores to suppress the low-quality detections. Instead of adding an extra branch, IoU-balanced classification loss [32] and PISA [33] adopt the normalized IoU and the sorted score, respectively, to reweight the classification loss based on their localization accuracy, which strengthens the correlation of the classification confidence and the localization quality. Besides, several works aim to design a joint representation of localization accuracy and classification. VarifocalNet [28] and generalized focal loss [29] modify the paradigm of focal loss and use IoU between bounding boxes and ground truth boxes instead of the target labels corresponding to the category of the positive samples. The IoU-aware methods naturally transfer classification scores to localization confidence, but the pure classification confidence is discarded, which is fatal for tasks like diagnosing tumors.

**Boundary Prediction.** To predict a specific form of instance, such as an instance mask and boundary, a conventional method is to attach an extra head to predict the task-specific outputs parallel to another task like classification and regression branches. YOLACT [34], as the first real-time approach method for instance segmentation, which is based on the RetinaNet [16], attaches a mask prediction head parallel to the classification and localization tasks, generating a set of prototype masks and then predicting per-instance mask coefficients. Similarly, CondInst [35] performs instance segmentation task by reformulating FCOS [17] where a lightweight mask map is predicted from the lowest level of FPN, while conditional convolution kernels are dynamically generated to integrate the mask map and obtain instance-wise masks. PolarMask [36] uses polar representation to model the outer contour of instance and transforms the per-pixel mask prediction problem to the distance regression problem. Our BGS also attaches an extra head to predict the boundary of each instance, but our target is to guide detectors to pay more attention to contour information for more precise localization. It is worth noting that we dismantle the prediction head during inference so that there is no extra computation for us to establish a densely boundary pixels prediction instead of using the quantitative forecasting manners [36,37].

## 3. Stacked Carton Dataset

The popular datasets [8,9,10,11,38] have been adopted to verify the performance of deep networks and have made great contributions to the development of computer vision. Regrettably, these benchmarks do not include carton scenes that are crucial for the application of logistics, transportation, and robotics. In this section, we introduce a new large-scale Stacked Carton Dataset (SCD) focusing on detection and the semantic understanding of cartons, which is suitable for various carton distribution scenarios and which has several appealing properties. First, we introduce the details of image collection and split, and then image annotation is described. Finally, the dataset statistics and properties are illustrated.

### 3.1. Data Collection

The source of data collection includes two aspects: online downloading and on-site shooting. In terms of images on the web, it is easy to find a large number images of carton because boxes are widely used in daily life, logistics, and other scenarios. Concretely, we integrated a batch of seed keywords and photos for matching and downloading; 8401 images were collected after removing duplicates and filtering. However, data collected from the web have a low resolution and even are mixed with noise such as watermarks and text. To mimic a real-world application scenario, we assembled 7735 images with high pixels and resolution from some typical carton stacking scenes, e.g., an E-commerce warehouse, a large wholesale market, integrated logistics warehousing, and a fruit market. There are several principles during data collection:The distance between the camera and stacked carton targets is controlled within 5 m.The layout of different cartons in one image appears within the same distance scale. Try not to let cartons far away and extremely close appear on the same horizon.One photo has only one group of cartons. If several piles of cartons appear, the distance scale between the target and the camera is strictly limited.Try to shoot from a perspective perpendicular to a group of cartons.

Compared with ImageNet [10], the source of our data is wider, the number of the image is several times more than it is in terms of carton class, and the number of instances in one photo is distributed from sparse to dense while one photo in the ImageNet benchmark only contains one or few cartons. Moreover, we give more meaningful labels for an instance to cater to other tasks described in Section 3.3.

### 3.2. Data Splits

To adapt to different tasks of diverse scenarios, we split the images into two subsets (called Live Stacked Carton Dataset (LSCD) and Online Stacked Carton Dataset (OSCD)), which all contain training and testing sets:LSCD: It contains all images taken at the carton stacking loading and unloading scene. The images were randomly split into two parts including 6735 images for training and 1000 images for testing.OSCD: It contains all images collected from the Internet to provide a substantial number of images for pre-training or transfer learning tasks. Likewise, 1000 images were randomly selected for testing, while the remaining 7401 images were for training.

Although only two subsets are specified, there are several different combinations. For example, LSCD and OSCD are integrated for pre-training or the semantic knowledge achieved from OSCD is transferred to related carton tasks.

### 3.3. Image Annotation

The same as with MS COCO [8], we used the polygon box to mark object instances. Figure 1 shows the pixel-wise annotations applied in LSCD (first line) and OSCD (second line), respectively. We defined only one annotation label (Carton) for annotation for images collected from the Internet and four labels (Carton-inner-all, Carton-inner-occlusion, Carton-outer-all, and Carton-outer-occlusion) for images in LSCD. One label (Carton) is also supported in LSCD so that it is easy to combine with OSCD for training and validation. We used the LabelMe project [39] to implement these annotations and transfer this labeling information into MS COCO [8] and PASCAL VOC [9] paradigms.

These are several rules and corresponding properties for the design of these four labels:**How to distinguish between inner and outer.** For a given carton in pile of cartons, the “inner” label is marked if the contour of instance is all contacted by cartons. The instance is truncated by image edges and is also judged to be connected to the cartons. All cartons except the “inner” label are “outer.” As shown in the first line of Figure 1, blue and green masks represent the “inner,” while red and yellow ones belong to the “outer.”**How to distinguish between all and occlusion.** Given a carton, as long as there is a complete face, the instance belongs to the “all” label, while the instance that has no complete face is assigned the “occlusion” label and similarly when a surface is truncated by image edges that have a deficient face. As the first line of Figure 1 shows, blue and red masks represent “all,” while green and yellow ones belong to “occlusion.”

### 3.4. Dataset Statistics

In Table 1, our dataset is compared with existing popular datasets containing related carton images. It demonstrates that ImageNet [10] and Open Image [11] only have a small number of carton images. In the Open Image dataset, only the label of “box” is provided, which contains part of the sub-data of the carton. Therefore, it cannot be utilized for related carton tasks. Our carton dataset SCD includes a total number of 16,136 images collected from the site and the Internet, which makes the images in SCD more diverse. Furthermore, topology information for the placement of the carton is provided in labels, which is important for some logistics scenarios. Finally, SCD is suitable for both the detection task and the segmentation task. By comparison, SCD has the advantages of a substantial quantity, rich annotated information, and superior quality. Thus, SCD is suitable for diverse tasks that are related to carton detection and segmentation.

Figure 2 and Figure 3 describe the relationship among the number of images, instances, and the four types of labels in terms of LSCD. The number of instances and images with respect to the “Carton-outer-all” label is more than that of the other labels. Figure 4 illustrates the distribution of instances in all images with respect to LSCD and OSCD, which contains the width, height, aspect ratio, and pixel area of a marked rectangular box. Width and height are normalized by the width and height of the corresponding images, respectively. The Log function is adopted to normalize the aspect ratio because it is boundless. The first four histograms of the two rows indicate that the width, height, and area distribution is close to the Gaussian model, and most instances have a medium size. In Figure 4a, the reason why the spike appears near the horizontal coordinate 1 is that a few slender cartons on LSCD are specially collected. Finally, (e) and (f) represent the number of cartons in each image, which demonstrates that the dataset obeys from a sparse to a dense distribution, and the maximum number of instances in one image is over 100, which is a challenge for instance detection and segmentation.

### 3.5. Property

As shown in Appendix A at the end, some raw images of ImageNet with respect to carton and our SCD are shown by random sampling. By comparison, it highlights the properties of LSCD, which are suitable for logistics scene. We list some corresponding property between ImageNet and LSCD (for simplicity, A represents the properties of ImageNet, while B is responsible for LSCD):**Distribution of instances**. A: instances almost distributed in the center of the image. B: The cartons are distributed in arbitrary positions of the photo.**Resolution**. A: blurred and low resolution. B: clear and high resolution.**Noise**. A: all images are downloaded from the Internet so that some noises, such as a watermark are attached. B: all images are collected at the scene and with almost no noise so that it can mimic the real scene well.**Texture**. A: mostly brown cartons and a white background. B: both the cartons and the background have rich textures.**purity**. A: it mixes with a carton-like box, such as wood, foam, and iron boxes, and some carton are squashed and wrinkled. B: it only contains cartons that are unfolded into boxes.

## 4. Methodology

Comparing with several CNN-based models, RetinaNet [16] was selected for detection because of its better trade-off between speed and performance. Actually, we expected that FCOS [17] will achieve better performance, but the result was the opposite (see Table 2). We speculated that the tricks that were designed for FCOS may not work in a simple scene, such as detection tasks with only one foreground category and background; so, we leave it to the researchers to explore in the future. In this section, we first review the RetinaNet and then introduce the novel Offset Prediction between Classification and Localization (OPCL), which is a unified and lightweight head attached to RetinaNet. Finally, Boundary Guided Supervision (BGS) is described to cure chronic diseases of carton detection and further boost the performance of the model.

### 4.1. RetinaNet

RetinaNet is a representative single-shot detector, as shown in Figure 5, whose network consists of a backbone and a feature pyramid network (FPN) with five detection heads. FPN is constructed by taking the backbone network with levels from $C_{3}$ to $C_{5}$ ($C_{i}$ is the output of corresponding ResNet residual stage) as input and fusing features map with a top-down pathway and a lateral connection. Each level of the pyramid is used for detecting objects at different respective scales. The multi-scale feature maps are tiled with densely hand-made anchors, which gives the detector the ability to predict various size instances. Its classification and localization subnets are both small fully convolutional networks consisting of four stacked convolution layers attached to $P_{l}$ ($P_{l}$ represents the output of the corresponding *ℓ*-th layer of FPN). The classification head predicts the probability of objects with respect to each anchor for each of the *K* object classes, while the localization head predicts the four-dimensional class-agnostic offsets between the anchor and the ground-truth box. Owing to the imbalance between hard and easy examples and the imbalance between foreground and background class, focal loss was adopted to ease this problem. The localization task uses Smooth $\ell_{1}$ function to regress the four offsets of anchors.

### 4.2. Offset Prediction between Classification and Localization (OPCL)

Vanilla RetinaNet only predicts the classification score and position for all bounding boxes. The classification task of each anchor is supervised by one-hot vector, which guided the detector only to learn a category confidence but may not contain localization confidence composition. Therefore, the anchor equipped with a larger classification score did not represent the positioning well. However, RetinaNet greedily takes the classification score as input to carry out the NMS procedure. In which case, the same boxes with high classification scores but low positioning quality may be retained, and the boxes with low classification scores but high positioning quality may be retained. This issue significantly damages the performance by as much as 14.5% [26]. Instead of directly predicting the localization confidence of each box, we explored to predict the gap from the classification score to the localization confidence.

**Learning to predict offset.** To bridge the gap from the classification score to the localization confidence, we need to define a metric for the localization confidence. Inspired by the evaluation manner of MS COCO [8], we adopted the Intersection over Union (IoU) between prediction boxes and ground-truth boxes to measure the localization confidence. Given a prediction classification score, how to fill the gap to achieve accuracy localization confidence is the key of OPCL. We first built a bridge between the two counterparts by Equation (Equation 1):(1)$$\hat{P_{i}^{loc}}=Sigmoid\left(C_{i}^{cls}+\hat{C_{i}^{off}}\right)$$
where $\hat{P_{i}^{loc}}$, $C_{i}^{cls}$, and $\hat{C_{i}^{off}}$ represent the localization confidence (i.e., the IoU between prediction box and corresponding ground-truth box), the prediction classification *activation*, and the corresponding *offset activation*, respectively.

To make an accurate estimate of localization confidence, it needs to predict the offset activation $C_{i}^{off}$. To do this, we selected to construct the offset prediction layer parallel with the classification layer. Actually, the regression head can also be attached, but we find that it obtains a suboptimal performance. The style of concatenation between the parallel heads were also implemented, but the performance was the same as the first one. Therefore, we chose the first style because of the negligible computation and better performance. As shown in Figure 5, a 3 × 3 convolutional layer with C × 1 channel was applied to output the offset prediction per spatial location.

**Training and inference.** The training and inference pipelines were shown in Figure 6. The binary cross-entropy loss (BCE) was adopted for offset prediction, whose performance was better than using other loss functions, i.e., L1 loss or L2 loss. Equations (Equation 2) and (Equation 3) show the loss functions of offset prediction. For the i-th positive anchor, $P_{i}^{loc}$ represents the prediction localization confidence; $C_{i}^{off}$ represents the offset before applying sigmoid function; and $C_{i}^{cls}$ represents the classification score before applying sigmoid function. During training, we added $C_{i}^{off}$ with corresponding $C_{i}^{cls}$ and then induced Sigmoid activation to achieve the prediction localization confidence. In $L_{OPCL}$, we used the $\hat{IoU}$ between the prediction box and the corresponding ground-truth box as a target to supervise the BCE loss. Similar to regression loss, only positive samples were applied to predict the offset. It is worth noting that whether to calculate the gradient of $L_{OPCL}$ with respective to IoU obtains completely different results. By transferring the gradient of IoU backward, the localization accuracy is further improved (results can be shown in Table 3).
(2)$$P_{i}^{loc}=Sigmoid\left(C_{i}^{cls}+C_{i}^{off}\right)$$
(3)$$L_{OPCL}=\frac{1}{N_{pos}}\sum_{i\in N_{pos}}^{}BCE\left(Sigmoid\left(P_{i}^{loc}\right),\hat{{IoU}_{i}}\right)$$

During inference, the predicted offset was adopted to compensate for classification scores. Equation (Equation 4) shows that the offset before activation was added to the classification score before and then fed into the sigmoid function, which is actually the predicted localization confidence. However, we did not use synthesized confidence as a criterion to apply the NMS procedure; instead, we defined a controlling factor α to integrate the original classification score and corresponding predicted localization confidence as $S_{i}^{det}$, which is the final criterion in the NMS procedure. Concretely, we conducted an element-wise multiply operation between classification scores and corresponding localization confidences. α was used to control the contribution of the two counterparts. The larger α, the greater the contribution of the localization confidence.
(4)$$S_{i}^{det}=Sigmoid{\left(C_{i}^{cls}+C_{i}^{off}\right)}^{\alpha }\cdot Sigmoid{\left(C_{i}^{cls}\right)}^{\left(1-\alpha \right)}$$

### 4.3. Boundary Guided Supervision (BGS)

As shown in (a) of Figure 7, stacked cartons often have repetitive texture information like the red dotted boxes. In this case, the boundary may be regarded as a section texture on a carton instance, which confuses the model to make the wrong positioning decisions. To solve this issue, it is indispensable to endow the model with the ability to identify boundaries among cartons with highly similar textures. To this end, we designed the BGS module to guide the detector to pay more attention to the precise boundaries of cartons, which is conducive to decoupling the highly repetitive textures and further promoting instances localization.

**Architecture.** BGS is a simple, unified network (we used a fully convolutional network (FCN)), targeting to predict precise boundaries of all carton instances in an image. As illustrated in Figure 4, we chose the first output layer of the feature pyramid network (FPN) to attach the BGS module because it equips with a better trade-off between high resolution and rich semantic information for better predicting boundary clues. The detailed architecture of BGS is shown in Figure 8, which is the same as the architecture of the classification and regression head. We firstly used four stacked 3 × 3 convolutional layers with C × C channels followed by ReLU to receive the selected FPN feature map and then adopted a 3 × 3 convolutional layer with C × 1 channels followed by sigmoid activations to predict the final boundary. Channel C was set as 256. Since the BGS model was built on the first layer of FPN, the prediction boundary result was 1/8 resolution of the input image.

**Loss and supervision.** We formulated the prediction boundary as a binary feature map with one channel. As Figure 7 shows, the corresponding boundary targets consist of binary values, in which the boundary pixels’ value is 1, while the other pixels’ is 0. To build the BGS loss, there are many choices for the loss of BGS, such as Binary Cross Entropy (BCE), Focal Loss (FL) [16], or Dice Loss (DL) [40]. How to choose a better loss depends on its effect on the performance improvements from supervision. Due to the natural imbalance between boundary and other pixels in prediction scenarios, where the pixels of boundaries always occupy few parts, FL and DL may be more suitable for the implementation of BGS loss. On our following experiments, the upper bound of FL is verified to be more advantageous than that of the other two losses mentioned above. To align with the output size of the prediction boundary, the target boundary will be down-sampled to 1/8 the resolution of the input image using bilinear interpolation. To do this, the thickness of the boundary needs to be carefully designed. Too small of a boundary thickness may lead to inadequate or no supervision, while a too large thickness may lead to vague supervision. We designed a parameter σ to control the boundary thickness so that the target pixels ${\left\{N^{t}\right\}}_{t=1}^{n}$ can be divided to ${\left\{N_{bdr}^{t}\right\}}_{t=1}^{n_{i}}$ and ${\left\{N_{nbdr}^{t}\right\}}_{t=1}^{n_{j}}$ for boundary pixels and non-boundary pixels, respectively. $n_{i}+n_{j}$ is equal to the total number of pixels *n* where σ control the number of $n_{i}$. The final BGS loss is:(5)$$L_{BGS}=\frac{1}{N_{bdr}}\sum_{i\in N}^{}\left(-\alpha_{t}^{i}{\left(1-p_{t}^{i}\right)}^{\gamma }\log\left(p_{t}^{i}\right)\right)$$
(6)$$p_{t}^{i}=\left\{\begin{array}{c} \begin{array}{cc} p^{i}, & i\in N_{bdr} \end{array}\\ \begin{array}{cc} 1-p^{i}, & i\in N_{nbdr} \end{array} \end{array}\right.$$
(7)$$\alpha_{t}=\left\{\begin{array}{c} \begin{array}{cc} \alpha , & i\in N_{bdr} \end{array}\\ \begin{array}{cc} 1-\alpha , & i\in N_{nbdr} \end{array} \end{array}\right.$$

The loss of $L_{BGS}$ calculates both boundary and non-boundary samples, but only the number of boundary samples is used for its weight loss. $p_{i}$ is the pixel value of the prediction boundary map. $\alpha_{t}$ is used to balance boundary and non-boundary pixels.

Note that the network of BGS is *removed* during inference so that there is no extra computation overhead. Finally, our detector’s total loss function is:(8)$$L_{total}=L_{cls}+L_{reg}+\gamma L_{OPCL}+\beta L_{GBS}$$
where the classification and regression loss weight is the same as the default detector, while γ and β is the weight of gap and boundary supervision task, respectively. We simply set γ and β to 1.

## 5. Experimental Results

In this section, several state-of-the-art detectors such as RetinaNet [16], Faster R-CNN [15], and FCOS [17] are trained on SCD as baselines. Then, our proposed modules (OPCL and BGS) are attached to the RetinaNet to demonstrate their superiority. To show the generalization ability of OPCL, several tables are stated, while two popular datasets (MS COCO and PASCAL VOC) are evaluated too.

### 5.1. Experiment Setting

**Dataset:** We evaluated the performance of our vanilla model and designed ablation studies on our dataset SCD and other challenging datasets including PASCAL VOC and MS COCO.

*SCD*: The dataset was split into LSCD and OSCD, which are formed by live photos and Internet photos, respectively. LSCD contains 6636 training and 1000 testing images, while OSCD includes 7401 and 1000 images for training and testing, respectively. In addition, LSCD supports two different experiments because of its two types of labels.

*PASCAL VOC 2007/2012:* The Pascal Visual Object Classes (VOC) [9] benchmark is one of the most widely used datasets for classification, object detection, and semantic segmentation. PASCAL VOC 2007 consists of 5011 images for training (*VOC2007 trainval*) and 4952 images for testing (*VOC2007 test*). PASCAL VOC 2012 consists of 11,540 images for training (*VOC2012 trainval*) and 10,591 images for testing (*VOC2012 test*). There are totally 20 categories of objects, and all these objects have been annotated with bounding boxes.

*MS COCO:* Recently, almost all the detection models take Microsoft Common Objects in Context (MS-COCO) [8] for image captioning, recognition, detection, and segmentation testing. It consists of 118k images for training (*train-2017*), 5k images for validation (*val-2017*), 20k images for testing (*test-dev*), and totally over 500k annotated object instances from 80 categories.

**Evaluation Protocol.** In this study, we adopted the same performance measurement as the MS COCO Challenge [10] to report our results. This includes the calculation of mean Average Precision (mAP) over different class labels for a specific value of IoU threshold in order to determine true positives and false positives. The main performance measurement used in this benchmark is shown by **AP**, which is averaged mAP across different values of IoU thresholds, i.e., IoU={0.5,0.55,⋯,0.95}. Then, $AP_{S}$ (AP for small scales), $AP_{M}$ (AP for medium scales), and $AP_{L}$ (AP for large scales) were evaluated.

**Implementation Details.** All experiments were implemented based on PyTorch and MMDetection [41]. ResNet-50, ResNet-101, and ResNeXt-101 were used as backbones in RetinaNet [16], which are pre-trained in ImageNet. A mini-batch of four images per GPU was used during training RetinaNet and other detectors, thus making a total mini-batch of eight images on two GPUs. The synchronized Stochastic Gradient Descent (SGD) was used for network optimization. The weight decay of 0.0001 and the momentum of 0.9 were adopted. A linear scaling rule was carried out to set the learning rate during training (0.005 in RetinaNet), and a linear warm-up strategy was adopted in the first 500 iterations. Except that the learning rate changes linearly with mini-Batch, all parameter settings are consistent with the default settings of MMDetection.

### 5.2. Baselines and Comparisons

**Baselines on Stacked Carton Dataset.** We report the results and comparisons of three state-of-the-art methods including RetinaNet, FCOS, and Faster R-CNN on SCD. As shown in Table 2, RetinaNet with GIoU loss (five is given for the weight of localization loss) performed best in all subsets of SCD with one or four labels. By comparison, RetinaNet is a detector with the best trade-off between performance and speed. Thus, RetinaNet was selected as the baseline to evaluate our novel modules and implement ablation studies. By pre-training all detectors in OSCD and then fine-tuning in LSCD with one or four labels, the performance of detectors can be significantly improved by 1.4∼7.3% AP, which indicates that the dataset collected from the Internet is effective to provide transcendental knowledge for carton detection in different scenarios.

### 5.3. Ablation Studies of OPCL on SCD

**The effectiveness factor of OPCL on different backbones. **Table 3 shows the performance of the Offsets Prediction between Classification and Localization module (OPCL), which is embedded into RetinaNet on the LSCD. OPCL with different backbones had a considerable increase of 3.1–4.7% over the baseline in mAP. It is worth noting that whether to apply backpropagation on the gradient of target IoU can make a significant performance difference. By reverse-passing the IoU gradient, it further improves the performance of OPCL by 1.8–3.3%.

**Which head to attach OPCL.** To investigate which head is suitable to attach OPCL, we constructed it followed by the classification head, teh regression head, and the combined head of the former two, respectively. The first two styles use the final feature map of the four stacked maps before the prediction results. For the style of the combined one, the feature maps of the former two styles are concatenated, and then we applied a 3 × 3 conv layer to obtain final offsets prediction. As shown in Table 4, OPCL that is attached on the classification head obtained a better result and costed fewer parameters than the combined style.

**Ablation of the control factor of OPCL.** After predicting the offsets of the classification score and the localization accuracy, the important step was to integrate the offsets to the classification score for inference. The control factor α was set from 0 to 1.0 to fine-tune the vanilla criterion, which was fed into the NMS procedure to implement a better ranking. As Figure 9 shows, yellow and gray polyline describe the change of performance with respect to OPCL with closing and an opening gradient backpropagation. It indicates that the control factor α has an optimal interval while having an optimal value in PASCAL VOC and MS COCO.

**Generalization Experiments on PASCAL VOC 2007/2012 and MS COCO.** The behavior of OPCL in PASCAL VOC is shown in Table 5, when the gradient of OPCL with respect to IoU was not computed during training; OPCL equipped with different backbones improved AP by 1.0∼2.4%. When the gradient of OPCL with respect to IoU was opened during training, the AP was improved by 3.4∼4.5%, while the AP at the higher IoU threshold (0.8 and 0.9) was improved by 9.1∼9.3%. The significant improvement about OPCL indicates that samples with worse localization accuracy were further improved in quality. As shown in Table 6, the conclusions from the experimental results of the MS COCO dataset are consistent with those from the experimental results of the PASCAL VOC dataset, which demonstrates OPCL has a generalization ability to other datasets and can be applied to different application scenes.

### 5.4. Ablation Studies of BGS on SCD

**Loss Function of BGS.** As Figure 10 shows, binary cross entropy loss, dice loss [40], and focal loss [16] were applied to learn the boundary map. As described in Section 4.3, the boundary maps prediction suffers from an imbalance between positive and negative samples so that result shows better performance with respect to focal loss. It is worth noting that the BGS module mainly improves the accuracy of high AP (${\mathrm{AP}}_{85}$, ${\mathrm{AP}}_{90}$, ${\mathrm{AP}}_{95}$ improve 1.2%, 2.6%, and 4.7%, respectively), which indicates that the BGS plays a key role in improving localization capability.

The use of focal loss brings two hyper-parameters, where the focusing parameter γ controls the strength of the modulating term, while the balance factor α controls the ratio of positive and negative samples in the loss. We adopted the same strategy as RetinaNet to search the great hyper-parameters. With γ=0.5 and α=0.5, FL yielded a 1.1 AP improvement over the BCE loss and the Dice loss.

**Boundary Thick of BGS.** One of the most important design factors in BGS is how to design the boundary thickness to supervise the loss function. Boundary information for each instance was extracted from the corresponding mask. The stride of 0.5 unit boundary thickness was adopted to search for the optimal supervisory thickness. Because the BGS module was attached on $P_{3}$, which reduces the resolution of the original image by eight times so that the actual stride of thickness is four pixels. As shown in Table 7, a series of isometric boundary thicknesses were searched, which surprisingly indicates that the mAP fluctuates less than 0.3 between pixel thickness 8 and 96. It shows that the boundary thickness has little effect on the performance. Theoretically, the minimum thickness should be greater than eight, which prevents the boundary from disappearing due to down-sampling, while too large of a thickness brings too much noisy information. However, deep works show great performance in anti-interference.

**Comparison results with other boundary styles.** We report the comparison results with other similarity boundary predictions. RetinaNet and YOLOX-M were employed as base detectors to carry out three manners of boundary supervision including our proposed BGS, PolarMask [36], and DeepSnake [37] manners. To be fair, we attached an extra branch parallel to the classification and regression branches during training. During inference, the boundary prediction heads were dismantled to keep the inference speed. The implementation of PolarMask [36] and DeepSnake [37] was the same as their official versions. As shown in Table 8, three boundary supervision manners all improved the performance of base detection, but BGS achieved the best improvement. We inferred that BGS monitors each pixel of the boundary for better boundary uncoupling, while PolarMask and DeepSnake only monitored the quantized boundaries, leading to ambiguous uncoupling. Although quantized boundaries supervision help achieved a higher inference speed on the instance segmentation task, we only employed the prediction manner to guide the model to pay more attention to the boundary region during training and threw away the corresponding structure in the inference stage. Thus, we can be greedy to choose the highest performance method without worrying about the speed burden.

### 5.5. All Modules on SCD

As shown in Table 9, all proposed modules were added to RetinaNet to make a stronger detector. The BGS module supervises the model to pay more attention to boundary information, which strengthens the ability of localization and improves performance to 80.9%. The OPCL module further improves a significant performance to 85.2% by bridging the gap of classification and localization quality. All modules improved baseline significant performance by 5.4% without any increase in computation during inference. Finally, we transfered the model parameter from the same detector pre-trained on OSCD, which further boosted 1.4% in mAP.

### 5.6. Main Results on Detection Task

Table 10 shows the main results of RetineNet with the OPCL, BGS, and pretraining strategy, which established a strong baseline. In addition, other advanced detectors were also compared. Our proposed OPCL and BGS endowed RetinaNet with comparable performance with the state-of-the-art detectors like YOLOX [20], DETR [19], and CenterNet [21].

### 5.7. Main Results on Instance Segmentation Task

We reported the performance of several instance segmentation models [18,22,23,24,25,34,42] on our SCD dataset, which provides a benchmark for the research on the carton instance segmentation. As shown in Table 11, HTC [25] achieved the best performance on both the detection and instance segmentation tasks. SOLOv2 showed an impressive performance on both segmentation accuracy and inference speed, which is suitable for the application.

## 6. Conclusions and Future Work

SCD is the first public dataset focusing on various carton layout scenes. The SCD dataset consists of two subsets with up to 16,136 images, which were collected from the Internet and diverse realistic scenarios. The number of instances in the images was distributed across sparse and dense scenes. The two subsets of SCD (LSCD and OSCD) simulated industrial carton stacking scenes and various generalized scenes in reality, respectively. OSCD downloaded from the Internet provides a priority in any carton-wise scenarios. SCD is annotated accurately at the pixel-wise level to be employed by detection and the instance segmentation using deep learning works. Based on SCD, several state-of-the-art models were evaluated. RetinaNet was selected to be plugged with OPCL and BGS, which balances classification and localization quality and guides the model to pay more attention to boundary information, respectively. We believe that SCD can provide a powerful application for the carton work scene and a new challenge to the computer vision community.

In the future, we plan to expand and refine various carton scenes and provide more labels for specific tasks, i.e., 3D detection, carton trademark detection, carton counting, transfer learning, and data augmentation, etc.

## Figures and Tables

**Figure 1 sensors-22-03617-f001:**
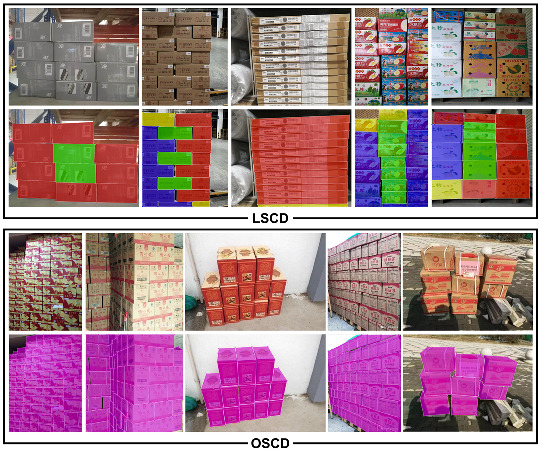
Example of instance annotation in SCD. The first first box represents the style of four labels with respect to LSCD, while the second box illustrates the style of one label in OSCD. In terms of the first box, blue, green, red, and yellow represent Carton-inner-occlusion, Carton-inner-all, Carton-outer-all, and Carton-outer-occlusion, respectively. In terms of the second box, the OSCD only contains the “Carton” label which uses purple masks.

**Figure 2 sensors-22-03617-f002:**
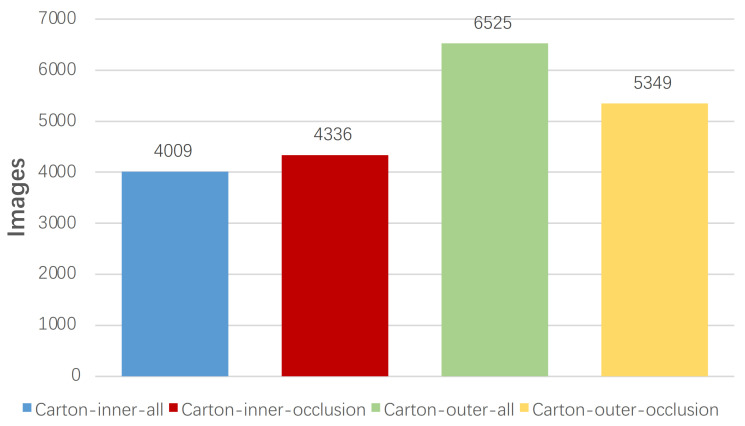
The numbers of instances with respective to four labels in the LSCD, which includes Carton-inner-all, Carton-inner-occlusion, Carton-outer-all, and Carton-outer-occlusion.

**Figure 3 sensors-22-03617-f003:**
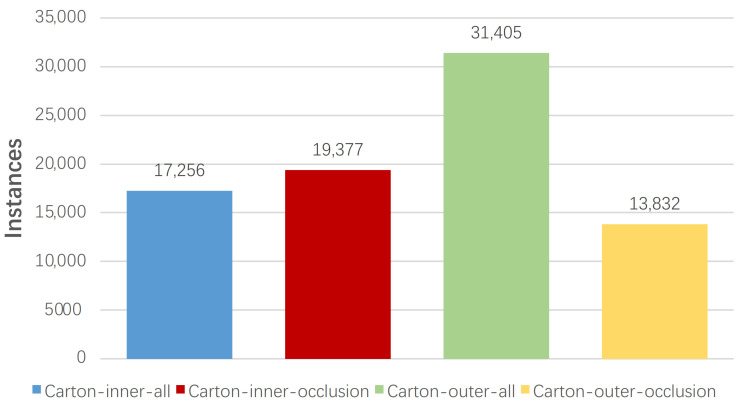
The numbers of images with respective to four labels in the LSCD, which includes Carton-inner-all, Carton-inner-occlusion, Carton-outer-all, and Carton-outer-occlusion.

**Figure 4 sensors-22-03617-f004:**
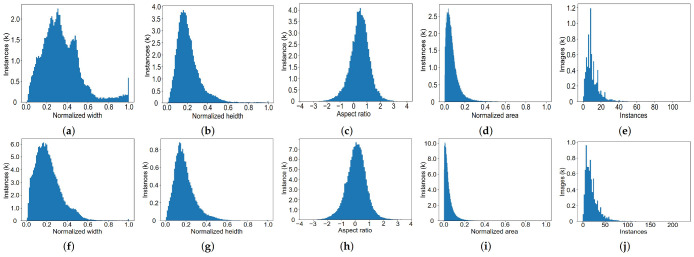
The first line (**a**–**e**) represents the statistical distribution of LSCD, while the second line (**f**–**j**) represents the statistical distribution of OSCD. The chart calculates the width, height, aspect ratio, pixel area, and the number of objects in each image from left to right. Note that the width, height, and area of instance are all normalized by the width and height of the corresponding image. The log function is adopted to normalize the aspect ratio.

**Figure 5 sensors-22-03617-f005:**
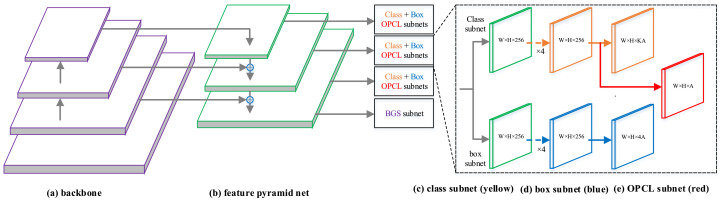
Illustration of our detector, which contains the backbone, feature pyramid net (FPN), and prediction head. The prediction head includes classification, regression, and Offsets Prediction between the Classification and localization (OPCL) head. The Boundary Guided Supervision module (BGS) is only attached at $P_{3}$ of FPN.

**Figure 6 sensors-22-03617-f006:**
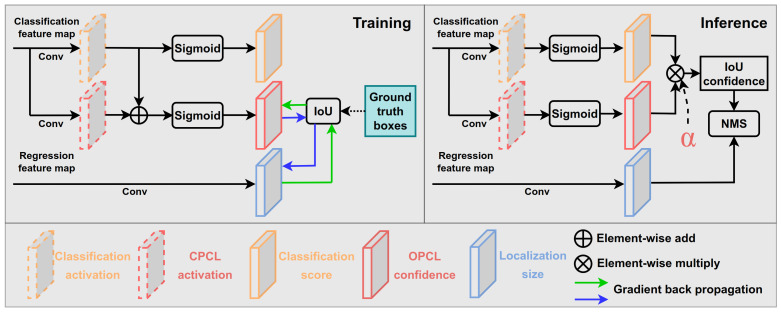
Illustration of the OPCL module, which contains training and inference pipelines. The classification and regression feature maps were achieved by conducting four stacked convolution layers with BN and ReLU. The classification and OPCL branches were attached on the same feature map.

**Figure 7 sensors-22-03617-f007:**
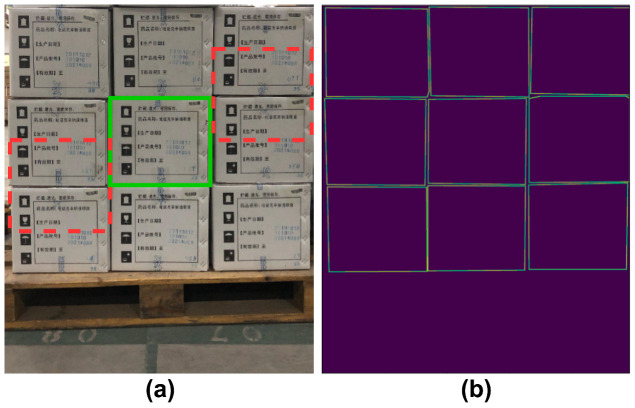
Illustration of the target boundary map that is extracted naturally from pixel-wise annotation. Green and red boxes in (**a**) represent correctly positioned boxes and error boxes that are confused by repeated texture. (**b**) is the binary image where the pixel value of the contour is 1.

**Figure 8 sensors-22-03617-f008:**
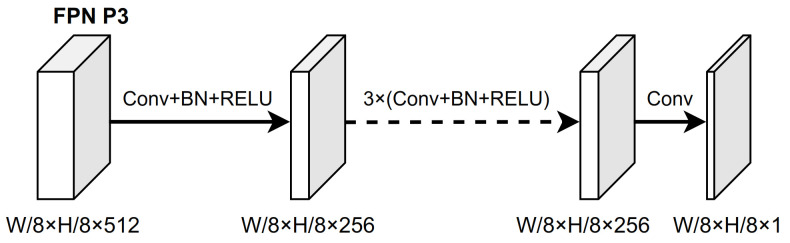
Illustration of the architecture of BGS head. W and H represent the origin width and the length of the input image, respectively. The feature map size is displayed as (width × depth × channel). Arrows indicate stacked 3 × 3 conv layer, while the final map represents the predicted boundary map whose size is W/8 × H/8.

**Figure 9 sensors-22-03617-f009:**
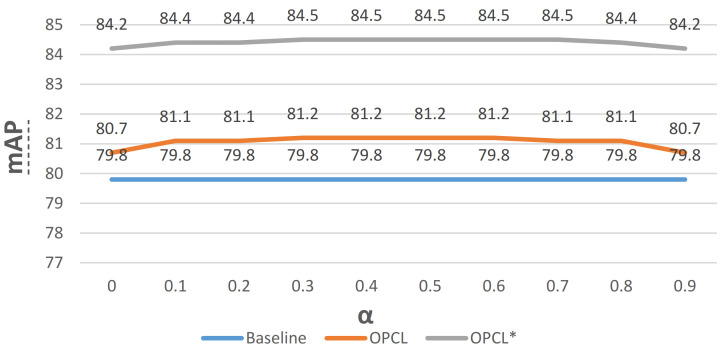
The searching process about control factor α on OPCL. The symbol “*” means the gradient of OPCL with respect to IoU is computed during training.

**Figure 10 sensors-22-03617-f010:**
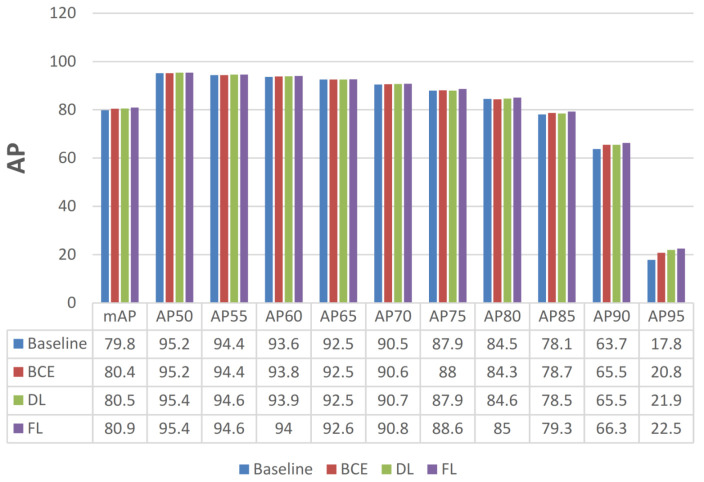
Comparison of three type of losses to Boundary Guided Supervision module (BGS).

**Table 1 sensors-22-03617-t001:** The comparison of datasets including the category of carton. “Category” represents the number of carton class. “4&1” in the LSCD column represents that it supports all four categories (Carton-inner-all, Carton-inner-occlusion, Carton-outer-all, and Carton-outer-occlusion) and one category (Carton) manners. Only LSCD supports occlusion information and inner/outer information by giving related annotations. “Images” means the number of images containing carton. “-” means the corresponding dataset only includes the category of box where carton only appear in images but the exact labels are not given in Open Image with too much noise for ImageNet to calculate the number of instances. “Average Instances” represents the average number of instances in each image. The checkmark represents the dataset contains the corresponding annotation while cross mark represents it does not contain.

Dataset	Images	Category	All/Occlusion	Inner/Outer	Carton Label	Total Instances	Average Instances
ImageNet [10]	1388	1	✗	✗	✓	-	-
Open Image [11]	-	1	✗	✗	✗	-	-
LSCD	7735	4&1	✓	✓	✓	81,870	10.58
OSCD	8401	1	✗	✗	✓	168,748	20.09

**Table 2 sensors-22-03617-t002:** Comparison of detection performance between three state-of-the-art methods on SCD. For the evaluation of LSCD, 1 and 4 labels are all evaluated. LSCD + OSCD means the detectors are firstly pre-trained in OSCD and then finetuned in LSCD. RetinaNet+ represents that GIoU loss was used.

Dataset	Labels	Model	mAP	${\mathrm{AP}}_{50}$	${\mathrm{AP}}_{75}$
OSCD	1	RetinaNet	72.1	90.8	80.5
OSCD	1	RetinaNet+	**76.6**	**91.8**	**83.6**
OSCD	1	FCOS	72.8	91.1	80.6
OSCD	1	Faster R-CNN	69.0	90.1	77.8
LSCD	1	RetinaNet	79.8	95.2	87.9
LSCD	1	RetinaNet+	**84.7**	**95.8**	**89.8**
LSCD	1	FCOS	76.5	93.7	84.3
LSCD	1	Faster R-CNN	77.5	94.5	86.3
LSCD	4	RetinaNet	65.7	80.4	73.0
LSCD	4	RetinaNet+	**69.9**	80.0	**74.9**
LSCD	4	FCOS	68.1	**81.2**	74.8
LSCD	4	Faster R-CNN	61.2	79.5	70.1
LSCD + OSCD	1	RetinaNet	82.2	95.9	89.8
LSCD + OSCD	1	RetinaNet+	**86.1**	**96.3**	**91.2**
LSCD + OSCD	1	FCOS	83.8	96.2	90.4
LSCD + OSCD	1	Faster R-CNN	80.6	95.7	89.2
LSCD + OSCD	4	RetinaNet	67.4	80.8	74.1
LSCD + OSCD	4	RetinaNet+	**71.5**	80.9	76.4
LSCD + OSCD	4	FCOS	71.1	**82.0**	**76.8**
LSCD + OSCD	4	Faster R-CNN	64.7	81.2	73.7

**Table 3 sensors-22-03617-t003:** Experimental results on LSCD. All the models are trained on LSCD trainval and evaluated on LSCD testing with the image scale of [600, 1000]. All the other settings are adopted as the same as the default settings provided in the MMDetection. The symbol “*” means that the gradient of OPCL with respect to IoU is computed during training.

Head	Backbone	mAP	${\mathrm{AP}}_{50}$	${\mathrm{AP}}_{60}$	${\mathrm{AP}}_{70}$	${\mathrm{AP}}_{80}$	${\mathrm{AP}}_{90}$
Baseline	ResNet-50-FPN	79.8	95.2	93.6	90.5	84.5	63.7
Baseline	ResNet-101-FPN	81.6	95.7	94.5	91.8	86.1	67.4
Baseline	ResNeXt-32x4d-101-FPN	82.1	96.0	94.7	92.0	86.6	67.9
OPCL	ResNet-50-FPN	81.2	95.0	93.8	91.0	85.2	67.0
OPCL	ResNet-101-FPN	82.9	96.0	94.8	92.2	87.2	70.1
OPCL	ResNeXt-32x4d-101-FPN	83.6	96.2	95.2	92.6	87.6	71.3
OPCL *	ResNet-50-FPN	84.5	95.3	94.0	91.7	87.1	74.5
OPCL *	ResNet-101-FPN	84.7	95.7	94.5	92.0	87.5	74.5
OPCL *	ResNeXt-32x4d-101-FPN	86.5	96.4	95.3	93.4	89.4	77.8

**Table 4 sensors-22-03617-t004:** Comparison between three types of head to attach OPCL.

Head	mAP	${\mathrm{AP}}_{50}$	${\mathrm{AP}}_{60}$	${\mathrm{AP}}_{70}$	${\mathrm{AP}}_{80}$	${\mathrm{AP}}_{90}$
cls	**84.5**	95.3	94.0	91.7	87.1	74.5
reg	84.1	95.4	94.2	91.6	96.6	73.9
cls&reg	84.4	95.4	94.0	91.6	87.3	74.5

**Table 5 sensors-22-03617-t005:** Experimental results on PASCAL VOC. All the models were trained on VOC2007 trainval and VOC2012 trainval and evaluated on VOC2007 test with the image scale of [600, 1000]. All the other settings were adopted as the same as the default settings provided in the MMDetection. The symbol “*” means the gradient of OPCL with respect to IoU was computed during training.

Head	Backbone	mAP	${\mathrm{AP}}_{50}$	${\mathrm{AP}}_{60}$	${\mathrm{AP}}_{70}$	${\mathrm{AP}}_{80}$	${\mathrm{AP}}_{90}$
Baseline	ResNet-50-FPN	51.4	79.1	74.6	64.0	45.4	15.9
Baseline	ResNet-101-FPN	55.1	81.1	77.2	67.5	50.4	20.1
Baseline	ResNeXt-32x4d-101-FPN	56.1	81.9	78.1	68.1	52.0	21.4
OPCL	ResNet-50-FPN	53.8	79.9	76.3	66.5	48.6	18.5
OPCL	ResNet-101-FPN	56.1	81.4	77.9	68.9	52.0	20.8
OPCL	ResNeXt-32x4d-101-FPN	57.9	82.6	79.4	70.0	54.9	23.6
OPCL *	ResNet-50-FPN	55.7	79.2	75.5	67.0	51.3	25.2
OPCL *	ResNet-101-FPN	58.5	80.2	77.0	69.6	55.8	29.2
OPCL *	ResNeXt-32x4d-101-FPN	59.6	78.3	70.7	70.7	57.0	30.7

**Table 6 sensors-22-03617-t006:** Experimental results on MS COCO. All the models were trained on COCO trainval and evaluated on COCO val-2017 with the image scale of [800, 1333]. All the other settings were adopted as the same as the default settings provided in the MMDetection. The symbol “*” means the gradient of OPCL with respect to IoU was computed during training.

Head	Backbone	mAP	${\mathrm{AP}}_{50}$	${\mathrm{AP}}_{60}$	${\mathrm{AP}}_{70}$	${\mathrm{AP}}_{80}$	${\mathrm{AP}}_{90}$	${\mathrm{AP}}_{S}$	${\mathrm{AP}}_{M}$	${\mathrm{AP}}_{L}$
baseline	ResNet-18-FPN	30.8	49.6	45.0	37.6	26.1	8.6	16.1	34.0	40.7
baseline	ResNet-50-FPN	35.6	55.5	51.0	43.2	31.1	11.3	20.0	39.6	46.8
baseline	ResNet-101-FPN	37.7	57.5	53.3	46.0	33.7	13.0	21.1	42.2	49.5
baseline	ResNeXt-32x4d-101-FPN	39.0	59.4	55.2	47.6	34.9	14.1	22.6	43.4	50.9
OPCL	ResNet-18-FPN	32.0	48.8	45.3	39.1	29.0	10.7	16.8	34.6	43.0
OPCL	ResNet-50-FPN	36.5	54.8	51.1	44.6	33.1	13.1	20.5	40.2	48.4
OPCL	ResNet-101-FPN	38.7	57.2	53.5	47.2	35.9	14.6	21.6	43.0	51.8
OPCL	ResNeXt-32x4d-101-FPN	40.4	60.0	55.9	49.2	37.4	15.8	22.9	45.1	53.3
OPCL *	ResNet-18-FPN	32.8	47.7	44.6	39.3	30.6	14.7	16.9	35.6	44.6
OPCL *	ResNet-50-FPN	37.4	54.1	50.4	44.8	35.0	16.8	20.8	41.4	50.0
OPCL *	ResNet-101-FPN	39.6	56.6	53.1	47.2	37.2	18.7	21.8	44.0	53.7
OPCL *	ResNeXt-32x4d-101-FPN	41.2	58.5	54.9	48.9	38.9	20.1	23.7	45.8	54.8

**Table 7 sensors-22-03617-t007:** Ablation performance of the thick with respect to BGS. Thick means the boundary pixel thickness on the input image.

Thick	8	16	20	24	28	32	36	40
mAP	80.6	80.6	80.6	80.8	80.7	80.7	80.8	**80.9**
**Thick**	**44**	**48**	**52**	**56**	**60**	**64**	**72**	**96**
mAP	80.8	80.7	**80.9**	80.7	80.8	80.8	80.8	80.8

**Table 8 sensors-22-03617-t008:** Comparison results on SCD among three types of boundary prediction manners. ResNet-50 was used as the backbone for RetinaNet with the image scale of [600, 1000]. YOLOX employed the default image size ([640, 640]).

Detector	Boundary	mAP	${\mathrm{mAP}}_{50}$	${\mathrm{mAP}}_{75}$
RetinaNet	BGS	**80.9 (+1.1)**	**95.2**	**87.9**
	PolarMask [36]	80.5 (+0.7)	95.3	88.0
	DeepSnake [37]	80.2 (+0.4)	95.2	87.8
YOLOX-M	BGS	**89.8 (+1.0)**	**97.4**	**94.1**
	PolarMask [36]	89.6 (+0.8)	97.4	93.7
	DeepSnake [37]	89.0 (+0.2)	97.0	93.0

**Table 9 sensors-22-03617-t009:** The performance of RetinaNet sequentially adding our proposed modules. All experiments adopt ResNet50 as a backbone with the image scale of [600, 1000]. The symbol “*” means the gradient of OPCL with respect to IoU was computed during training.

Baseline	✓	✓	✓	✓	✓
BGS		✓	✓	✓	✓
OPCL			✓	✓	✓
OPCL *				✓	✓
pre-training					✓
mAP	79.8	80.9	82.1	85.2	**86.7**

**Table 10 sensors-22-03617-t010:** Main results of RetinaNet with all our proposed modules and other advanced detectors. “Pretrain” means pretraining identity model on OSCD and fine-tuning on LSCD with the image scale of [600, 1000] (“†” means the image scale is [800, 1333]). “‡” means fine-tuned from COCO dataset model. “1x” means the model was trained for total 12 epochs. The bold font is used to emphasize the best performance according to the comments of the reviewers. In ${\mathrm{AP}}_{S}$ item, this was on the corresponding value because SCD are on small instances in the evaluation framework of MS COCO. The symbol “*” means the gradient of OPCL with respect to IoU was computed during training.

Detector	Backbone	Schedule	mAP	${\mathrm{AP}}_{50}$	${\mathrm{AP}}_{75}$	${\mathrm{AP}}_{S}$	${\mathrm{AP}}_{M}$	${\mathrm{AP}}_{L}$
DETR	ResNet-50	150 epoch	70.9	92.0	80.6	-	34.6	71.2
DETR	ResNet-50	300 epoch	77.6	94.3	86.3	-	37.8	77.9
YOLOX-M	Modified CSPDarknet-53	300 epoch	71.8	92.4	81.7	-	34.9	72.3
YOLOX-M ‡	Modified CSPDarknet-53	1x	88.8	96.7	93.0	-	55.2	89.0
CenterNet	ResNet-50	1x	86.4	96.4	91.2	-	50.0	86.6
OPCL * + BGS	ResNet-50-FPN	1x	85.2	96.0	90.1	-	48.2	85.3
OPCL * + BGS	ResNet-101-FPN	1x	86.0	96.1	90.6	-	46.9	86.3
OPCL * + BGS	ResNeXt-32x4d-101-FPN	1x	86.7	96.4	91.6	-	49.0	86.9
OPCL * + BGS + pretrain	ResNet-50-FPN	1x	86.7	96.5	91.5	-	51.1	87.0
OPCL * + BGS + pretrain	ResNet-101-FPN	1x	87.5	96.5	92.4	-	50.0	87.7
OPCL * + BGS + pretrain	ResNeXt-32x4d-101-FPN	1x	87.5	96.5	92.2	-	48.1	87.8
OPCL * + BGS + pretrain †	ResNet-50-FPN	1x	87.5	96.6	92.0	-	48.8	87.8
OPCL * + BGS + pretrain †	ResNet-101-FPN	1x	87.6	96.6	92.5	-	52.0	87.8
OPCL * + BGS + pretrain †	ResNeXt-32x4d-101-FPN	1x	**89.0**	**97.2**	**93.7**	-	**55.5**	**89.2**

**Table 11 sensors-22-03617-t011:** Instance segmentation benchmark on SCD. All models were trained with the image scale of [800, 1333]. We used ResNet-50 pretrained on ImageNet dataset as a backbone for all models.

Method	Schedule	${\mathrm{mAP}}^{box}$	${\mathrm{AP}}_{50}^{box}$	${\mathrm{AP}}_{75}^{box}$	${\mathrm{mAP}}^{seg}$	${\mathrm{AP}}_{50}^{seg}$	${\mathrm{AP}}_{75}^{seg}$
Mask R-CNN [18]	1x	85.2	96.1	91.8	88.0	96.1	93.1
BlendMask [22]	1x	86.1	96.2	92.0	88.5	96.3	93.3
Cascade Mask R-CNN [42]	1x	87.6	96.6	92.1	89.0	**96.6**	93.7
HTC [25]	1x	**88.2**	**96.7**	**93.0**	**89.6**	**96.6**	**94.0**
YOLACT [34]	1x	66.6	90.6	77.8	71.7	90.5	80.4
SOLO v1 [23]	1x	-	-	-	79.0	92.9	86.2
SOLO v2 [24]	1x	84.3	95.7	91.0	88.9	96.2	93.6

## Data Availability

We published our dataset at https://github.com/yancie-yjr/scd.github.io, accessed on 9 March 2021.

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
