# Peer review of "SCD: A Stacked Carton Dataset for Detection and Segmentation"

_sensors, 2022, doi:10.3390/s22103617_

Round 1

Reviewer 1 Report

This is an interesting paper about the detection of stacked carton which is very useful and practical for the industrial applications.

The presentation style of the paper is well designed and easily understandable for reader.

The authors created the large-scale carton dataset of Stacked Carton Dataset (SCD) and performed the experiments, and analyse the results.

Moreover, The authors also conducted the extensive experiments on the public datasets of MS COCO and PASCAL VOC for confirming the generalization of proposed model.

The results are also reasonable.

Therefore, I agree to publish this paper.

Author Response

We thank you for taking time out of your busy schedule to review and approve our work!

Reviewer 2 Report

This paper offers a dataset for large-scale carton scene detection. In addition, it provides a baseline method that incorporates two proposed modules into RetinaNet. However, the motivation for improving RetinaNet is not clear. There are many expressions that are incorrect.  Some equations are incorrectly used. Overall, the manuscript needs careful revision. Detailed comments are listed below:

  • I suggest the title be changed to “SCD: A dataset for …” as the manuscript provides a new dataset.
  • Line 7: “Per-instance segmentation”->”instance segmentation”
  • Line 58: “RetinaNet suffers from a serious imbalance [19,20] between classification and localization accuracy.” is not substantiated. The author should explain this problem in the introduction as it is the motivation for the two proposed modules. I don’t understand this sentence “In other words, the result of classification scores and localization quality is not equal absolutely and their relationship is even nonlinear.” 
  • All: leave a space between text and citation, eg. text  [cite]
  • Line 143 “Map and Boundary Prediction”: what does “Map” mean? In addition, instance mask is not a “target feature map”.
  • Figure 1: the annotation color almost covers the original picture, it is hard to differentiate the four types of cartons. In this case, the original images should be placed together.
  • Table1: the quotations should be kept the same for the whole manuscript. "Labels" and ’4&1’ are not acceptable.
  • Fig 7: why correct vs. error use red and green color, the two colors should be switched to avoid misleading information.
  • Line 300: C^gap is unclear.
  • Eq 2: BCE is for classification, how can it be applied to regression? Eq3 is not explained.
  • Table 2: the best performance in each dataset should use bold.

Reviewer 3 Report

Authors label and propose new open-source dataset of Carton's for instant segmentation.
Authors proposes some modifications to RenitaNet to set the benchmark for Carton detection. Dataset and benchmarks are valuable. Resource for various engineering tasks, and practitioners.
Scientific novelty, however, is limited. 
In such, I would recommend publication due to a worth of large open dataset and quality benchmark for Carton detection.

General notes:

* Reviewer recommends to leave the name from pre-print: 
Scd: A stacked carton dataset for detection and segmentation
This will help to collect the citations and avoid multiple copies of same publication.* Text[Citation] -> Text [Citation]
* figure -> Figure
* Figure 4. - all images should be redrawn and saved as pdf/eps with 300+ dpi, and imported independently.
* Text(Text) -> Text (Text)
* ℓ-th level -> ℓ-th layer/transformation
* Figure 6 -> Add notation width x height x channels
* Equ -> Eq.
* NMS, add notation of non-maximal suppression in manuscript
* Eq 4. no index i, fix formula
* Eq 7 missing weights near each loss.

Author Response

We thank you for providing insightful comments. Our responses to the mentioned questions are below:

Response 1: We decided to change the title to “SCD: A Stacked Carton Dataset for Detection and Segmentation”, which will help to collect the citations and avoid multiple copies of same publication.

Response 2: We have further polished our manuscript including:

* Text[Citation] → Text [Citation]

* figure → Figure

* Text(Text) → Text (Text)

* Equ → Eq.

* ℓ-th level → ℓ-th layer/transformation

* Add notation of non-maximal suppression in manuscript when NMS is first mentioned.

* We have fixed the incorrect description of Figure 6, which is highlight with red font.

* We have fixed the incorrect formulas on Eq. 4 and Eq. 7 including adding index and loss weights.

* We have updated all figures by saving as PDF with 300+ dpi.

Reviewer 4 Report

The article is devoted to the actual practical topic of stacked carton scene detection. The authors provided the open source code for the proposed method and made the dataset publicly available, which is useful for the reproducibility of the obtained results.

At the same time, there are a number of comments on the work:

1) The authors should have compared the chosen detection approach based on the RetinaNet model with other popular and effective approaches: YOLOX, CenterNet, Transformer-based model DETR.

2) Since the labeling is made in the “pixel-wise” style with the selection of individual mask instances, it should be noted why there was no analysis of the effectiveness of instance segmentation models such as Mask R-CNN, HTC (Hybrid Task Cascade), BlendMask, SOLOv2, etc.

3) Separately, the article highlights the aspect of taking into account information about the contours during model learning. In connection with this, the authors should compare their approach with existing methods: DeepSnake [Peng, S., Jiang, W., Pi, H., Li, X., Bao, H., & Zhou, X. (2020). Deep snake for real-time instance segmentation. In Proceedings of the IEEE/CVF Conference on Computer Vision and Pattern Recognition (pp. 8533-8542)], PolarMask [Xie, E., Wang, W., Ding, M., Zhang, R., & Luo, P. (2021). Polarmask++: Enhanced polar representation for single-shot instance segmentation and beyond. IEEE Transactions on Pattern Analysis and Machine Intelligence.], etc.

4) The authors should describe in more detail the BGS subnet and OPCL subnet, which are a key elements of the novelty of the proposed model, since the basic RetinaNet is well known. It is desirable to add a diagram with the structure of these modules to the article and clearly describe the output data that these modules generate.

After eliminating these comments, the article is recommended for acceptance.

Round 2

Reviewer 2 Report

Thanks for the authors' effort to make this manuscript better. Some minor problem can be improved in the final verison, for example

Line 323 "(results can be shown in Table).", in which Table?

Author Response

We thank the Reviewer #2 for providing insightful comments. Our responses to the mentioned questions are below:

Point 1: Line 323 "(results can be shown in Table).", in which Table?

Response 1: We have added the corresponding table link.

Reviewer 4 Report

The authors corrected the main comments indicated in the review, which significantly improved the article.

But there were a number of small comments that should also be corrected in the new version of the article:

  1. In Figure 6. “Illustration of OPCL module…”, authors should clarify on the diagram what input data is used by the OPCL module. In the diagram, authors should replace "element multiply" and "element add" with "element-wise multiply" and "element-wise add"
  2. In formula (4), authors need to explain what kind of mathematical operation is between two Sigmoids
  3. In Figure 7, it is recommended to sign the arrows with the operations that they perform. The same is recommended for Figure 6.
  4. It is not clear on which dataset the metrics in Table 8 "Comparison results on boundary prediction" were obtained and it is not clear why Table 11 "Instance segmentation benchmark on SCD" does not contain the same methods indicated in Table 8.
  5. In Table 8, there is wrong refence for DeepSnake [36] - it should be [37].
